# CausalXRL: Explainable Reinforcement Learning through Causal Graph Reasoning

**Yanming Zhang** [1]  **Eric Papenhausen** [1]  **Klaus Mueller** [1]

## Abstract

Reinforcement learning is a powerful paradigm for training autonomous agents and has achieved impressive performance in complex environments. However, this success often comes at the cost of interpretability, diminishing trust and complicating efforts to debug and improve agent behavior. To address these challenges, we introduce CausalXRL, a novel framework for explainable reinforcement learning (XRL). A key feature of CausalXRL is its use of causal graph reasoning, which provides transparent, structured, multi-level explanations of agent decision-making. We validate CausalXRL through comprehensive case studies and a two-part evaluation: (1) a quantitative analysis of explanation fidelity and causal-structure learning efficiency in benchmark RL environments, and (2) a qualitative expert study assessing explainability in the real-time strategy (RTS) benchmark MicroRTS. The quantitative results show that CausalXRL can provide faithful explanations while efficiently learning causal structures, and the qualitative expert study suggests that participants found CausalXRL useful for inspecting high-level RTS strategies.

## 1. Introduction

Recent advances in machine learning (ML) have transformed many fields, with reinforcement learning (RL) emerging as a powerful but opaque paradigm (Puiutta & Veith, 2020). RL trains agents to select actions that maximize cumulative rewards in interactive environments, typically formalized as Markov Decision Processes (MDPs) where actions depend only on the current state. This makes RL well-suited for planning and sequential decision-making,

and it has been successfully applied in domains such as gaming, self-driving cars, robotics, and healthcare. However, traditional RL struggles in environments with high-dimensional inputs—such as pixels, sound, or large-scale sensor data—limiting generalization. Deep Reinforcement Learning (DRL) addresses these challenges by combining RL with deep neural networks, enabling agents to solve complex, real-world tasks. Notably, well-trained DRL agents have outperformed experienced human experts (Warchalski et al., 2020; Alicioglu & Sun, 2022) and even defeated top masters, as in AlphaGo's historic victory over Go champion Lee Sedol in 2016.

However, this power comes at a cost: DRL models are inherently opaque, making their internal reasoning difficult for humans to interpret or predict. This lack of transparency poses a barrier to adoption in high-stakes domains such as healthcare (Vallée, 2024). To address this, Explainable Reinforcement Learning (XRL) has emerged, aiming to provide clear, human-readable explanations of how and why agents make decisions. XRL research has largely focused on post-hoc explainability (Wells & Bednarz, 2021; Wang et al., 2022; Madumal et al., 2020; Qing et al., 2022; Dazeley et al., 2023) and can be broadly categorized into model-, reward-, state-, and task-level approaches (Qing et al., 2022). State-based methods clarify how a state arose (history), how it influences an agent's action (influence), and what it may lead to (intent), often using saliency maps to highlight input regions most relevant to decisions (Greydanus et al., 2018; Huber et al., 2019; Simonyan et al., 2013). By contrast, model- and reward-based approaches focus on the "why," explaining why an action was selected over alternatives and assisting developers with debugging (Dazeley et al., 2023), while some translate RL models into simpler proxy models for improved interpretability.

Providing causal explanations aligns with the mechanism how people make sense of the world (Sloman & Sloman, 2009). While it is not new to incorporate causality into reinforcement learning—many works have leveraged causality to guide the RL learning process (Wang et al., 2025; Seitzer et al., 2021; Cohen et al., 2020)—leveraging causal inference methods for explanation tasks in RL remains sparse. Madumal et al. (2020) employed graphical causal models

---

[1]Department of Computer Science, Stony Brook University, Stony Brook, USA. Correspondence to: Yanming Zhang <yanmzhang@cs.stonybrook.edu>.

*Proceedings of the 43rd International Conference on Machine Learning*, Seoul, South Korea. PMLR 306, 2026. Copyright 2026 by the author(s).

as an inherently interpretable approach to reasoning. They introduced the Action Influence Model (AIM), which extends structural causal models by explicitly incorporating the agent's action space into the causal edges. However, the construction of AIM requires a predefined causal structure from domain experts, and its action space is limited to finite sets. Yu et al. (2023) made the approach more practical by incorporating conditional independence tests (CITs) to determine the causal structure and introducing attention-based inference networks to compute the action contributions associated with each causal edge. This adaption enables AIM for continuous action spaces.

Yet, we identify two limitations. First, AIM's design, which binds causal edges directly to actions, makes it impractical for use in multi-unit environments such as real-time strategy (RTS) games. Because units are typically tied to specific capabilities by design, the resulting causal dependencies become deterministic rather than probabilistic. For example, in RTS games, a causal edge *Ally Attacking Units→Enemy Base* would always be present when ally attacking units are destroying the enemy's base, since this is the only capability tied to the attacking units. Humans derive limited additional insight beyond what can be obtained by simply watching the game, and in this case, AIM diminishes the ability of causal models to provide strategic or high-level explanations. Second, current structural learning algorithms that discover causal structures in the framework rely on CITs, which are extremely time-consuming. Their worst-case time complexity is $O(n^{k+2})$, where $k \gg 1$ is the maximum number of neighbors per node in the true graph, which constrains the efficiency of the entire XRL pipeline. In contrast, our proposed CausalXRL eliminates the need for predefined causal structures, avoids the computational burden of conditional independence tests, and generalizes beyond single-agent settings by operating over human-interpretable representations rather than binding causal edges directly to actions. Unlike AIM and its CIT-based extension, CausalXRL scales efficiently to multi-agent environments while remaining fully data-driven.

Our proposed CausalXRL is a new XRL framework designed to elucidate the decision-making process of model-free reinforcement learning agents through causal models. The pipeline of the framework can be divided into three segments: representation extraction, causal dependency inference and explainable causal models. In the representation extraction stage, we generate and efficiently collect interpretable representations from raw RL experiences. The resulting representation pool is then used to train Rubin Causal Models (1974) to infer causal dependencies. Causal structures can be efficiently inferred from these dependencies at time complexity $O(n^2)$. Lastly, these models are used to construct global causal graphs and local causal models, each serving complementary explanatory purposes. The global causal graphs provide an overview of the causal relationships among the extracted representations, while the local causal models offer situational insights that enable humans to perform counterfactual reasoning in specific contexts. We evaluated our approach on four RL benchmarks, and the results indicate that it accurately predicts behavior while remaining computationally efficient. Furthermore, we conducted an exploratory qualitative expert study using our tool in MicroRTS, a classic RTS game environment (Ontanon, 2013). Participants' feedback suggests that the framework has the potential to provide useful and interpretable explanations of agent strategies. Code can be found at: `https://github.com/yanmluk/causal-xrl`

## 2. Related Work

### 2.1. Interpretable Representations Extraction

Raw environment data in high-dimensional environments such as Atari games, MuJoCo, and RTS games are not directly understandable to humans. To support the downstream tasks of explanation generation, a common technique is to extract interpretable representations from the environment. At the state level, a popular method is to extract critical states or key decision points within the trajectories (Guo et al., 2021; Cheng et al., 2023). While this approach works well for visual tasks, it is less effective in environments with multiple agents or units, where humans can easily lose focus. In such cases, abstraction of the representations at different levels is required. (Topin & Veloso, 2019) introduced Abstracted Policy Graphs (APGs). An APG is a Markov chain built over abstract states, where each abstract state groups together many low-level environment states. Another indirect yet functional method is attribution, notable approaches such as LIME (Ribeiro et al., 2016), CLEAR (White & Garcez, 2019) and SHAP (Lundberg & Lee, 2017) can filter important features so that people can concentrate on them.

A recent breakthrough toward achieving human-level abstractions is the development of concept-based methods. These approaches, such as Concept Bottleneck Models (CBMs) (Koh et al., 2020), enforce a mapping from raw observations to human-interpretable concept features via a *concept encoder*, such that these concepts are sufficient for a *task predictor* to determine the agent's action decisions. In most existing formulations, the concepts are pre-specified by domain experts.

Prominent extensions of this paradigm include Successive Concept Bottleneck Agents (Delfosse et al., 2025), which learn relations between objects as concepts and thereby extend the definition of a concept beyond a property of a single object. Concept Bottleneck Policies (CBPs) (Grupen et al., 2022) adapt CBMs to multi-agent settings, where each agent's action is conditioned on a set of human-interpretable

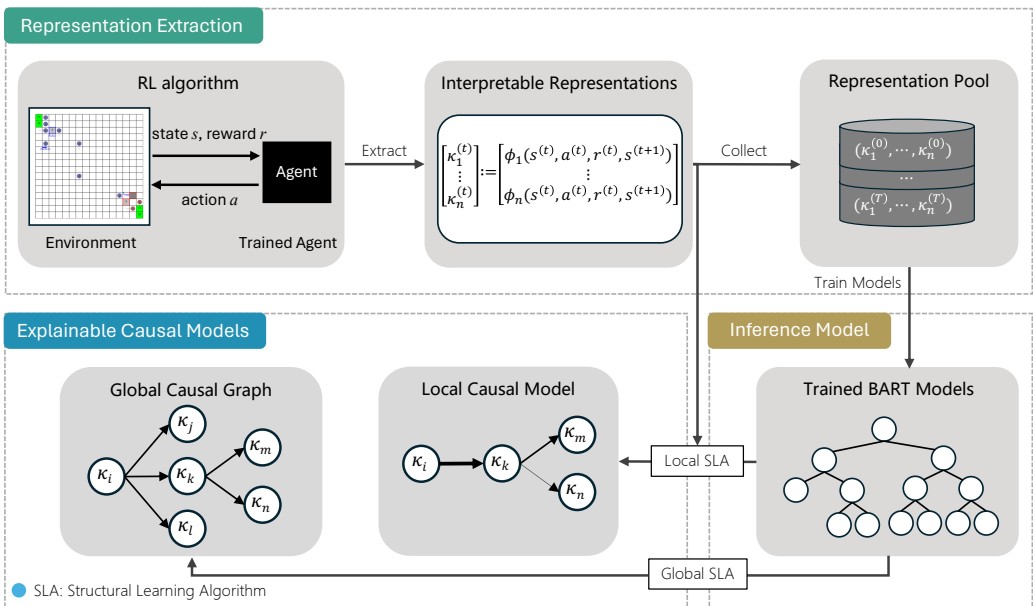

*Figure 1.* The architecture of CausalXRL. SLA is the Structural Learning Algorithm.

concepts. The concepts obtained from CBMs are human-understandable and represented as vectors, which makes causal discovery feasible in the subsequent step.

### 2.2. Causal Inference

Causal inference is a cornerstone of decision-making. It analyzes how a treatment variable directly influences an outcome, in contrast to correlation, which only captures co-movement. Correlation can be misleading when *confounders*, variables that affect both treatment and outcome, are present. Without proper adjustment for confounders, estimates of causal effects may be biased, resulting in spurious relationships that obscure the true underlying effects. Contrarily, an effect adjusted for confounders can support what-if analysis.

Unbiased estimation of causal effects is grounded in two foundational frameworks: Structural Causal Models (SCM) (Pearl, 2009) and the Rubin Causal Model (RCM) (Rubin, 1974). SCM emphasizes adjustment using graphical causal models, typically represented as directed acyclic graphs (DAGs), while RCM adopts the Potential Outcomes (PO) framework, relying on statistical methods to approximate randomized trials from observational data. Both models estimate causal or treatment effects by comparing potential outcomes under different treatments. In a classic binary treatment scenario, the causal effect can be formally defined as *average treatment effect* (ATE): $\mathbb{E}[Y(1) - Y(0)]$, where $Y(1)$ and $Y(0)$ represent potential outcomes under treated and control, respectively. In real-world scenarios that involve multilevel treatments or continuous treatments, treated and control value need to be further specified.

### 3. Background: MicroRTS

Gym-MicroRTS (2021), built upon MicroRTS (2013), is a multi-agent RTS environment designed for academic research in AI and game theory. Compared with more complex RTS environments such as StarCraft II, MicroRTS retains the core mechanics of RTS gameplay while significantly reducing the size and complexity of the state and action spaces. A typical game involves two opposing players, where each player is treated as an agent that centrally controls multiple units on its team. The objective of each player is to defeat the opponent by strategically managing resources, producing units, and eliminating enemy units.

Figure 2 summarizes the visual representation of each unit type and its capabilities. Unlike image-based environments, in which observations are commonly provided as RGB arrays, gym-MicroRTS represents observations using an integer-encoded feature space that explicitly captures unit-level information. In a game played on a $16 \times 16$ grid map, each cell corresponds to either an empty tile or a unit. Each unit is described by 27 discrete features, including unit type, team affiliation, carried resources, health, and current action. The features across all $16 \times 16$ grid cells together define the fully observable game state at a given time step. This state is represented as a $16 \times 16 \times 27$ tensor and is available to both players.

### 4. Methodology

Figure 1 provides an overview of CausalXRL, which comprises three modules: Representation Extraction, Inference Model, and Explainable Causal Models.

| Unit type | | Name | Capabilities |
|---|---|---|---|
| Ally units | Enemy units | | |
| | 25 | Resource | A Resource isn't owned by any player, can't perform actions. The number displays the remaining resources. |
| 5 | 5 | Base | A Base has 10 HP and costs 10 resources to build. It can only produce Workers. The number shows resources held by the player (ally/enemy). |
| | | Barrack | A Barrack has 4 HP and costs 5 resources to build. It can produce Light, Heavy, or Ranged units. |
| | | Worker | A Worker has 1 HP and costs 1 resource to build. It can carry 1 resource, move, attack (cause 1 damage), harvest, and return in adjacent cells. |
| | | Ranged | A Ranged has 1 HP and costs 2 resources to build. It can move to adjacent cells or attack locations with $d \leq 3$ (cause 1 damage). |
| | | Light | A Light has 4 HP and costs 2 resources to build. It can only move to or attack (cause 2 damage) adjacent cells. |
| | | Heavy | A Heavy has 4 HP and costs 2 resources to build. It can only move to or attack (cause 4 damage) adjacent cells. |
| Action type | —— move | —— attack | —— produce —— harvest or return |

*Figure 2.* Graphical representations of units and actions. HP (Hit Points): amount of damage a unit can sustain before destruction.

The Representation Extraction Module starts with the classic interaction between the Reinforcement Learning environment and the trained agent. For each step in the episodes, an experience tuple can be extracted, and then formulated into a list of interpretable representations. These representations are selectively collected through episodes and finally stacked to form a representation pool. Next, the Inference Model Module treats the data from the representation pool as observational data and employs the Rubin Causal Model (RCM) to uncover the causal effects between representations. The training concludes when it converges or reaches a specified number of iterations. The Explainable Causal Models module reveals the causal dependencies among representations to enhance human understanding. The Global Causal Graph uncovers the overall causal relationships between representations from replay episodes, while the Local Causal Model reflects the given experience with each directed edge parameterized indicating the causal strengths.

### 4.1. Representation Extraction

The raw features in complex environments often contain extensive information. Enabling humans to perform causal analysis in such environments generally requires human-interpretable concepts. While some RL environments include only a small set of key features that are understandable to humans, such as Bipedal Walker and Lunar Lander (Brockman et al., 2016), directly leveraging the raw feature space in more complex settings, such as RTS environments, is typically intractable for two reasons. First, it is highly space-inefficient due to the sparsity of the tensor representation, as raw features are often stored as images (e.g., Atari Pong) or maps (e.g., MicroRTS). Second, even when raw features are interpretable, some environments, such as MuJoCo Humanoid (Todorov et al., 2012), contain so many features that they overwhelm human users and

hinder understanding. To address these issues, we adopt a principled *representation extraction process* that simplifies complex raw data into a human-understandable format.

Numerous studies have proposed methods to extract interpretable concepts or representations from raw feature space at different levels of abstractions (Kim et al., 2018; Zabounidis et al., 2023; Ye et al., 2025; Delfosse et al., 2025). While some prior work (Kim et al., 2018) supports unsupervised or weakly supervised concept discovery—allowing models to automatically identify latent factors—substantial human involvement, such as grouping, pruning, and alignment, is still required to render these concepts. Therefore, we adopt expert-specified interpretable representations $\kappa^{(t)}$ derived from raw experience $e^{(t)} = (s^{(t)}, a^{(t)}, r^{(t)}, s^{(t+1)})$, defined as $\kappa^{(t)} = \Phi(e^{(t)})$. Here, $\Phi$ denotes a mapping from raw experience to interpretable representations. Depending on the modality of the raw state in the RL environment, such representations can be obtained using approaches such as concept encoders.

Specifically, in the presented MicroRTS environment, we employ high-level *strategic features* nominated by domain experts, where each feature represents the count of units of a specific type (e.g., number of ally Ranged units or enemy Barracks). These strategic feature values are computed via deterministic functions over the raw state features.

Once the interpretable representations are defined, we collect $n$ interpretable representations $\kappa^{(t)} = (\kappa_1^{(t)}, \ldots, \kappa_n^{(t)})$ by replaying multiple episodes, tracking the evolution of these representations, and storing them into the representation pool $D = \{\kappa^{(0)}, \ldots, \kappa^{(T)}\}$ such that $\kappa^{(t-1)} \neq \kappa^{(t)}$.

### 4.2. Causal Dependency Inference

Our approach focuses on dependencies among representations that can be represented as causal dependencies or graphically as directed edges. There are two main properties that distinguish a causal dependency from correlational dependency. 1) a causal dependency is directed. That is, a directed edge signifies that the change of the source variable can lead to the change of the target variable but not the other way around. 2) a causal dependency should adjust the distortion from third variables. The weight that associates with each directed edge should measure the change of target variable due to the source variable only. This requires controlling for the disturbance from other variables such as confounders.

Embedding these two properties in a causal inference setting, the source node is the treatment variable and the target node is the outcome variable. The weight of a directed edge should quantify the influence of treatment on outcome if the treatment was changing from control to treated. Here, we formulate the dependency using the Ru-

bin Causal Model (RCM). A RCM can be formulated as a tuple $\mathcal{M} = (X, Z, Y, f, h)$, $X$, $Z$, and $Y$ contain the variables from the representation pool, where $Z$ denotes the *treatment variable*, $Y$ denotes the *outcome variable*, and $X = \{X_1, X_2, ..., X_p\}$ is the set of *covariates* that might moderate the treatment effect. $f$ and $h$ are functional models where, depending the inference model we use, $f$ models the response surface and $h$ models the treatment assignment mechanism (Hill, 2011).

There are various powerful inference methods (Athey & Imbens, 2016; Wager & Athey, 2018; Künzel et al., 2019; Hill, 2011) that have been explored under the RCM framework. In this work, we use Bayesian Additive Regression Trees (BART) due to their flexibility, ability to capture nonlinear relationships, and built-in uncertainty quantification. BART models approximate an unknown function $f$ by representing it as a sum of many regression trees, where each individual tree acts as a weak learner in a boosting pipeline. Following the previous notation, the BART model can be written as,

$$Y = f(z, x) = \sum_j g(z, x; T_j, M_j) + \epsilon \qquad (1)$$

where $T_j$ is a tree, and $M_j = \{\mu_1, \mu_2, \ldots, and \mu_b\}$ represents the tree's terminal nodes with each node $\mu_i$ associated with a prediction; $g$ is the function that assigns the prediction to $(z, x)$, an instance of $(treatment, covariates)$ tuple; $\epsilon$ is a Gaussian noise term.

After properly fitting $h$ using existing methods (Rosenbaum & Rubin, 1983; Robins et al., 2000; Hill, 2011), we assume conditional ignorability—a standard assumption in causal inference for RL—since all relevant state variables are included in the representation pool. Then, the BART models can estimate the causal influence from the treatment $Z$ to the outcome $Y$ as the Conditional Average Treatment Effect (CATE):

$$\mathbb{E}[Y(1) - Y(0) \mid X = x] = f(1, x) - f(0, x) \qquad (2)$$

By applying the stored interpretable representations to this framework, we can hypothesize, test, and estimate the effect of a causal dependency $\kappa_i \rightarrow \kappa_j$. This process requires fitting a BART model with given treatment, outcome, and covariates tuples $(\kappa_i, \kappa_j, \kappa_{-\{i,j\}})$ such that $\kappa_{-\{i,j\}} = \{\kappa_1, ..., \kappa_n\} \setminus \{\kappa_i, \kappa_j\}$. Notably, our structure learning algorithm has a worst-case complexity of $O(n^2)$, which is substantially more scalable than CIT-based methods ($O(n^{k+2})$). This efficiency allows CausalXRL to handle environments with dozens of interpretable features, making it practical for real-world multi-agent scenarios.

---

**Algorithm 1** Global Causal Graph Structure Learning

**Input:** interpretable representations $D$.
**Output:** directed edges $\mathcal{E}$.

1: **/* Preprocessing */**
2: **for** $t = 0, 1, \ldots, T$ **do**
3: $\quad \Delta\kappa^{(t)} = \kappa^{(t+1)} - \kappa^{(t)}$       *// rep. increment*
4: $\quad \tilde{\kappa}^{(t)} = \left(z(\kappa_{n \in N}^{(t)}), \kappa_{n \notin N}^{(t)}\right)$    *// standardization*
5: **end for**

6: **/* Construct the causal graph */**
7: **for** $i = 1, 2, \ldots, n$ **do**
8: $\quad$ **for** $j = 1, 2, \ldots, n$ **do**
$\qquad \Delta\kappa_i = f(\tilde{\kappa}_j, \tilde{\kappa}_{-\{i,j\}})$
9: $\qquad\qquad = \sum_k g(\tilde{\kappa}_j, \tilde{\kappa}_{-\{i,j\}}; T_k, M_k) + \varepsilon$
10: $\qquad$ *// fit inference models (BART)*
11: $\qquad \tau_{ji}^{(t)} = f(\tilde{\kappa}_j(1), \tilde{\kappa}_{-\{i,j\}}) - f(\tilde{\kappa}_j(0), \tilde{\kappa}_{-\{i,j\}})$
12: $\qquad$ *// generate CATE samples*
13: $\qquad$ **if** $P_{2.5}(\tau_{ji}^{(t)}) > 0$ **or** $P_{97.5}(\tau_{ji}^{(t)}) < 0$ **then**
14: $\qquad\qquad$ Append causal link $(\kappa_j, \kappa_i)$ to the set $\mathcal{E}$
15: $\qquad$ **end if**
16: $\quad$ **end for**
17: **end for**

---

### 4.3. Explainable Causal Models

In order to generate human-understandable explanations considering the interrelationships among interpretable representations, we introduce two types of causal graphs: global causal graphs and local causal models.

Global causal graphs that contain all interpretable representations summarize causal relationships across all episodes and emphasize causal reasoning from an overall perspective. They help explore RL agents' overall strategies. In contrast, local causal models highlight context-aware reasoning and usually maintain only a subset of representations.

#### 4.3.1. GLOBAL CAUSAL GRAPH

Algorithm 1 outlines the global causal structure learning process. In short, the global causal graphs can be obtained by iterating through all interpretable representations as candidate treatment and outcome variables and computing their CATEs. To perform the procedure, we require the collected interpretable representations $D = \kappa^{(0)}, \ldots, \kappa^{(T)}$. Lines 1–5 describe the preprocessing procedure, where we compute increments of consecutive representations and perform z-score standardization only on numerical representations to improve model convergence. Lines 6–17 sketch the steps to construct a causal graph. Line 9 trains an inference model such as BART to predict the increment of representation $\kappa_i$. The resulting model is then used to compute CATE sam-

ples, with $\tilde{\kappa}_j(1)$ representing the treated value for $\kappa_j$ and $\tilde{\kappa}_j(0)$ the control value (Line 11). Without further specification, the default $(treated, control)$ value tuple equals $(1, 0)$. Lines 13–14 confirm a causal dependency if the ranked majority of CATE samples are either larger or smaller than zero. $P$ denotes the percentile.

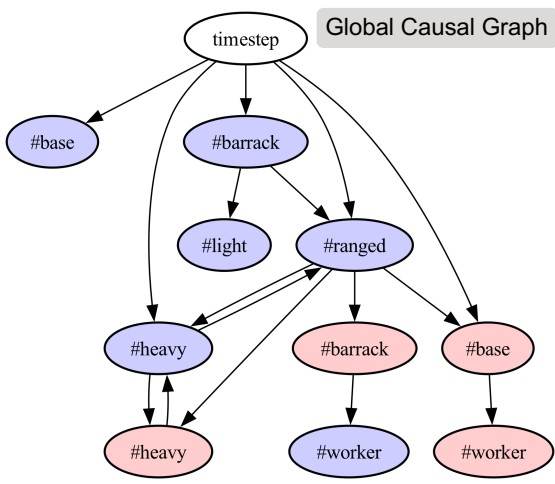

*Figure 3.* An example global causal graph revealing the overall causal dependencies among representations learned by a sample of trained RL agents in MicroRTS.

The demonstrated algorithm is a statistically conservative approach to causal discovery that aims to eliminate false positive dependencies. As a more flexible alternative, applying clustering algorithms such as agglomerative clustering to covariate–CATE pairs can partition the CATE samples into multiple clusters. A causal link is inferred if any resulting cluster exhibits a consistent non-zero effect. This approach accounts for heterogeneity in treatment effects, unveiling more subtle causal dependencies.

Figure 3 illustrates an example of a global causal graph learned from the MicroRTS environment. By viewing a global causal graph, analysts can instantly gain an overall impression of how representations are causally related. For instance, the directed edge *Blue Barrack $\rightarrow$ Blue Light* suggests that more Barracks can produce more Lights. To reduce confounding during training, *time step* — the elapsed game steps — is included as an additional feature. Without it, the model might learn spurious dependencies driven by patterns that co-occur at similar stages of gameplay. Although used during training, this variable and its associated edges are omitted from local causal model visualizations to avoid misinterpretation.

#### 4.3.2. LOCAL CAUSAL MODELS

Algorithm 2 outlines the construction of local causal models that condition on a particular experience and reuse the trained BART models to reveal situation-specific causal de-

---

**Algorithm 2** Local Causal Model Structure Learning

**Input:** a single step experience $e^{(t)}$, z-score scalers $z$, and trained BART models $\mathcal{F}$.
**Output:** a set of directed edges $\mathcal{E}$ parameterized with weights.

1: $\kappa^{(t)} = \Phi(e^{(t)})$ // Convert raw exp. into repr.
2: $\tilde{\kappa}^{(t)} = (z(\kappa_{n \in N}^t), \kappa_{n \notin N}^t)$ // standardization
3: **for** $i = 1, 2, \ldots, n$ **do**
4:     **for** $j = 1, 2, \ldots, n$ **do**
5:         $\tau_{ji}^{(t)} = f(\tilde{\kappa}_j(1), x_{-\{i,j\}}) - f(\tilde{\kappa}_j(0), x_{-\{i,j\}})$
6:         // generate CATE samples
7:         **if** $P_{2.5}(\tau_{ji}^{(t)}) > 0$ **or** $P_{97.5}(\tau_{ji}^{(t)}) < 0$ **then**
8:             Append causal link $(\kappa_j, \kappa_i, \bar{\tau}_{ji}^{(t)})$ to the set $\mathcal{E}$
9:         **end if**
10:     **end for**
11: **end for**

---

pendencies. As opposed to a global causal graph, which only allows one to reason about the source of change, a local causal model parameterized by numerical influence can help analysts determine how much tweaking is needed, and in what direction, to approach a target value for a representation of interest.

Figure 4(a-b) presents a local causal model derived from a MicroRTS game state. In this scenario (a), blue Workers are gaining momentum, with one Worker constructing a Barrack. Meanwhile, team red has prioritized Barrack construction and is producing a Heavy unit. The corresponding causal model in (b) shows that the number of red Heavy units is negatively influenced by both blue Ranged and Heavy units, while the number of blue Barracks positively affects the number of blue Ranged units. Conversely, the negative feedback loop between blue Heavy and Ranged units suggests that increasing one may suppress the other, indicating a trade-off in the agent's unit production priorities. However, the edge originating from blue Ranged units is slightly stronger ($-0.88$ vs. $-0.55$). From the agent's perspective, the causal chain *Blue Barrack $\rightarrow$ Blue Ranged $\rightarrow$ Red Heavy* therefore highlights a promising strategy for suppressing team red's strength. This strategy is reflected in the ensuing steps, where the outcome in (b) shows that a blue Ranged unit produced in the constructed Barrack eliminates a red Heavy unit.

Meanwhile, the local causal model enables counterfactual reasoning. In the same scenario, if the agent were to produce more blue Heavy units, the causal chain *Blue Heavy $\rightarrow$ Red Heavy* suggests that the number of red Heavy units would decrease. We simulated this intervention, and the strategy is later verified when a blue Heavy unit destroys a red Heavy unit.

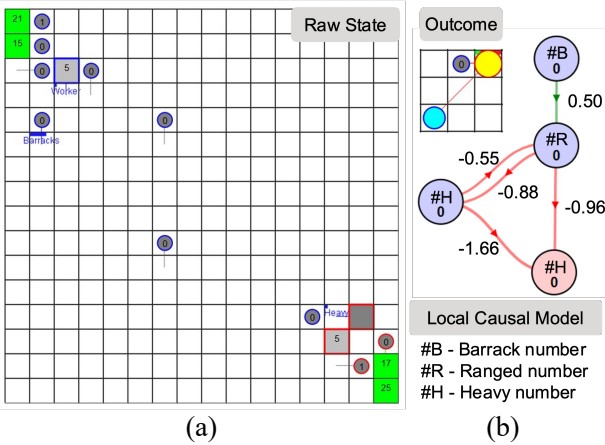

(a)           (b)

*Figure 4.* Examples of Explainable Causal Models in MicroRTS. (a) A game state sampled from an episode. (b) A local causal model revealing the RL agent's strategy. Blue nodes represent allied units, and red nodes represent enemy units. Directed edges indicate what-if dependencies, with green and red denoting positive and negative relationships. The partial outcome at the top left validates the causal model's predictions.

### 4.3.3. INTERPRETATION OF LEARNED CAUSAL GRAPHS

The learned causal models should be understood as human-interpretable causal surrogates of the trained policy's behavior. They are intended to be behaviorally faithful to the policy over collected trajectories, rather than to recover the agent's internal reasoning or neural computations.

## 5. Computational Evaluation

We evaluate the proposed explainable causal models in a range of single-agent (LunarLander and CartPole) and multi-agent environments (PongDuel and MicroRTS-Lite). The evaluation focuses on two primary objectives:

- **Behavior Prediction**: Determine whether the learned local causal models are faithful to the behaviors of trained RL agents;

- **Structural Learning Efficiency**: Examine whether the proposed structural learning algorithm for the global causal graphs are computationally more efficient than CIT based structural learning algorithms.

### 5.1. Experiment Design

We first execute trained RL agents for 200 episodes and store the resulting transitions in a replay buffer. These collected transitions serve as a hold-out dataset for training local causal models, where the interpretable representations consist of the original state features and the corresponding actions.

### 5.1.1. BEHAVIOR PREDICTION

The goal of behavior prediction is to assess whether the causal model faithfully reflects the agent's behavior in a given state. An intuitive proxy is to compare the actions inferred by the local causal model with the actual actions produced by the trained agent. A faithful local causal model should satisfy two criteria: 1) the parent nodes of the action node should contain sufficient information to accurately predict the action, and 2) the weights of the edges into the action node should reflect the predictive contribution of the corresponding parent nodes.

Since the local causal model infers values exclusively for endogenous variables, the test dataset was specifically filtered to include only experiences in which actions appeared as endogenous variables. For simplicity, action prediction is assumed to depend solely on its immediate parent variables in the local causal graph. For each parent variable, we train a BART model using the parent's data together with additional covariates that are not identified as causes according to the global causal graph. Each BART model produces an action prediction conditioned on a specific parent. The final predicted action is then obtained by a weighted aggregation of these predictions, where the weights correspond to the relative strengths of the local edges connecting each parent to the action node. The inference performance of our causal approach was compared against baseline methods, including linear models (LM) and decision trees (DT), both of which predict actions directly from state features without constraints imposed by the causal graph.

These experiments do not include AIM-style methods as baselines because their explanatory target differs from ours. AIM estimates the influence of individual actions on state transitions by associating causal edges with actions. In contrast, CausalXRL learns dependencies among strategic state representations. In RTS games such as MicroRTS, the effect of an edge may encode the influence of many simultaneous unit-level actions and interactions from both agents, making it ill-defined to map each strategic edge to a single action.

### 5.1.2. STRUCTURAL LEARNING EFFICIENCY

We conducted empirical runtime experiments on the structural learning algorithms. Specifically, we measured the time required by each algorithm to construct the global causal graph. Each algorithm was run five times, and we report the average runtime.

Since CIT-based structural learning algorithms typically produce more consistent causal graphs than score-based algorithms such as GES (Chickering, 2002), we compare our approach against the CIT-based baselines PC (Spirtes et al., 2001) and FCI (Spirtes et al., 1995).

| RL Env | #Rep | Accuracy (%) | | | F1-score (%) | | | Efficiency (s) | | |
|---|---|---|---|---|---|---|---|---|---|---|
| | | LM | DT | Our | LM | DT | Our | PC | FCI | Our |
| CartPole | 4 | 88.5 | **94.1** | 91.4 | 90.0 | **94.6** | 91.6 | 1.7 | 0.8 | **0.1** |
| LunarLander | 8 | 81.3 | 85.1 | **88.7** | 80.2 | 84.7 | **86.6** | 12.6 | 5.1 | **0.3** |
| PongDuel | 12 | 81.5 | 88.3 | **90.2** | 81.1 | 88.4 | **90.7** | 23.2 | 8.7 | **0.5** |
| MicroRTS-Lite | 14 | 75.3 | 80.3 | **86.0** | 74.5 | 82.2 | **85.9** | 20.9 | 7.4 | **0.5** |

*Table 1.* Experimental results for behavior prediction accuracy, F1-score, and structural learning efficiency.

## 5.2. Results

Table 1 summarizes behavior prediction performance, measured by *Accuracy* and *F1-score* (average='weighted' if multiclass), as well as the training time required to construct the global causal graph, reported as *Efficiency*. In terms of faithfulness, our model performs competitively with existing baselines in relatively simple environments, such as CartPole and LunarLander, while achieving substantially larger gains in more complex environments, such as Pong-Duel and MicroRTS. In terms of learning efficiency, the proposed framework shows a clear advantage over other CIT-based baselines, particularly in complex environments involving a larger number of representations (*#Rep*).

## 6. Human Experts Evaluation

We conducted an exploratory expert study to understand how RTS domain experts interpret and use CausalXRL. The objective of this study is twofold: 1) To assess whether the proposed explainable causal models enhance human understanding of the decision-making processes of RL agents in complex environments. This includes interpreting the interdependencies among high-level features and gaining insight into the agents' strategic planning. 2) To examine whether improved understanding of agent behavior fosters greater human trust, particularly in scenarios involving human–autonomy teaming. We asked participants to think aloud during the study in order to understand how they reason strategies and perceive strategies from provided explanations.

### 6.1. Experiment Setup

Centering on MicroRTS, for the experiment, we developed an interactive, browser-based interface (Figure 7, appendix). The backend stores the replay buffer containing experiences and the trained causal models. On the frontend, the interface visualizes states and actions to mimic the original environment, augmented with the corresponding local causal models. Participants can interact with a slider to navigate across different time steps, allowing them to explore various states and inspect the associated local causal models.

To reach qualitative conclusions (Nielsen & Landauer, 1993), we recruited 8 RTS game experts (E1–E8) via a graduate student email list, each with over 7 years of experience and averaging more than 1,000 hours playing various RTS games such as StarCraft II, Age of Empires II, and Age of Mythology. This sample size aligns with prior qualitative XRL studies, where domain expertise rather than quantity is critical for reaching insight saturation. The study protocol was approved by the Stony Brook University Office of Research Compliance (ORC) under IRB protocol Application No. 1020950. All participants provided informed consent before participating. Participants were informed of the study purpose, procedure, data usage, and their right to withdraw at any time. Participants received compensation of $8/hr.

### 6.2. Experiment Design

The sessions were conducted via Zoom, allowing participants to remotely control the interface and provide verbal feedback in real time. Initially, participants learned the rules of MicroRTS and the basics of causality, then completed a brief quiz to ensure understanding. If a participant answered any quiz question incorrectly, the experimenter reviewed the relevant concept with the participant before they retook the quiz. Participants proceeded to the main study only after answering all questions correctly within three attempts; otherwise, they were excluded from the analysis. In our study, all participants passed the quiz on their first attempt.

Similar to (Lim & Dey, 2009), in the first phase of the main study, participants watched a gameplay video clip and identified the causal dependencies they observed, writing down each dependency using the provided interpretable representations while verbalizing their reasoning. They were then presented with the global causal graph learned from the same gameplay and asked to compare those two.

Next, we conducted a controlled behavior-prediction experiment. Given a state, participants predicted what would happen after a few subsequent steps in a multiple-choice question format. Participants were randomly assigned to either a control or an experimental condition. In the control condition, they could navigate previous states and inspect feature-level SHAP values, but did not have access to local causal models. In the experimental condition, they could inspect local causal models, but could not navigate between states. We used a crossover design so that each participant completed different questions under both conditions.

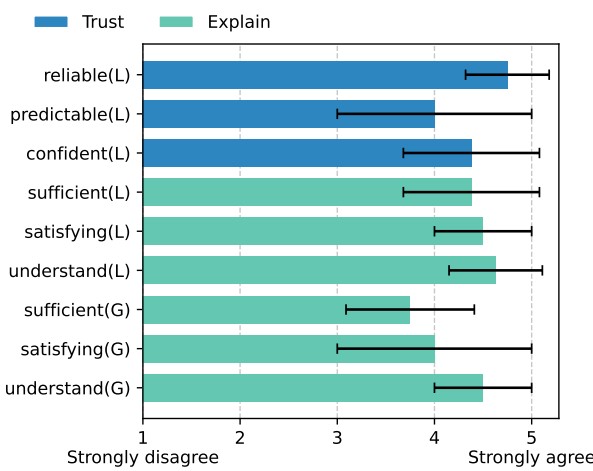

*Figure 5.* Survey ratings on CausalXRL's explanation quality and participants' trust: Global (G) and Local (L) Models.

Following the task, participants were asked to complete a Qualtrics questionnaire covering aspects of explainability and trust. Each question was rated on a 5-point Likert scale, ranging from 1 (Strongly disagree) to 5 (Strongly agree). In particular, we investigated Madumal's (2020) explanation metrics (understandability, satisfaction, and sufficiency) for global causal graphs and local causal models, as well as trust metrics (confidence, predictability, and reliability) for the local causal models.

### 6.3. Results

The participants' questionnaire responses are summarized in Figure 5. Additional details on the questionnaire items and participant responses are provided in Appendix B. The results reported in this section are intended to provide exploratory and suggestive evidence. [1]

**Support for learning agent strategies.** Participants described CausalXRL as accessible and useful for understanding the agent's strategic behavior. E2 noted, "*CausalXRL is an intuitive XRL framework where users can easily navigate through high-level explanations. I believe that beginners in RTS games can quickly catch on using the taxonomy CausalXRL provides.*" Several experts emphasized that the global causal graph helped reveal strategic patterns that would otherwise require extended observation. As E4 explained, "*CausalXRL captures the strategies of both ally and enemy sides through causal networks. Normally, discovering such strategies would require a significant amount of time and self-driven observation.*" This benefit was also reflected in participants' MCQ performance: participants with access to the local causal model achieved higher accuracy

---

[1] Due to the limited number of participants, the study was not designed for statistically powered quantitative comparisons, but rather to illustrate behavioral trends.

in behavior prediction than those in the control condition (8/8 vs. 4/8 correct).

**Understanding and actionable reasoning.** Participants generally rated the explanations from both global and local causal models positively, with most explanation metrics receiving average ratings above 4(Somewhat agree). However, feedback loops in CausalXRL models were not always immediately interpretable. For instance, E5 initially found the mutual negative feedback loops in Figure 4 difficult to understand. After clarification from the experimenter, E5 recognized the production tradeoff as clear and valid, noting, "*I need to compare the effect magnitude between Rangeds and Heavys to understand that the loop between the two nodes is a tradeoff rather than a contradiction.*" This feedback suggests that users may need guidance to translate graph structures into counterfactual reasoning and, ultimately, into gameplay implications.

**Gaining trust in predicting agent behavior.** High trust ratings were reported for local causal models' ability to capture agent behavior, with all trust-related metrics scoring above 4. As E8 stated, "*Compared with what SHAP provides, the local causal model offers directed links, which makes me feel more confident in predicting the agent's next moves.*"

## 7. Discussion and Conclusion

In this paper, we introduced CausalXRL, a post-hoc causal reasoning framework for model-free reinforcement learning agents. Our approach learns underlying mechanism between interpretable representations in complex environments, enabling it to generate insightful explanations through learned causal models. We evaluated CausalXRL in four benchmark RL domains, demonstrating its effectiveness in both behavior prediction and causal structure learning. We then conducted a qualitative user study with domain experts, focusing on behavior prediction, explainability, and trust. The expert study provides exploratory evidence that participants found CausalXRL useful for interpreting agent strategies and has the potential to enhance user's trust in RL agents.

While promising, CausalXRL still has several limitations. First, it relies on expert-specified interpretable representations, which may limit its applicability when suitable abstractions are unavailable. Second, scaling causal discovery and visualization to hundreds of variables remains challenging for industrial-scale deployments.

In future work, we plan to extend CausalXRL to support interactive alignment, enabling experts to edit causal graphs and simulate the resulting policy changes. This would transform explanations into tools for controllable agent behavior. These capabilities position CausalXRL as a foundation for human-in-the-loop alignment, enabling experts to interrogate, refine, and steer agent behavior.

## Impact Statement

This paper presents work whose goal is to advance the field of Machine Learning. There are many potential societal consequences of our work, none which we feel must be specifically highlighted here.

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

# Appendix

## A. Simulation Details

All environments discussed in the paper took place on a $16 \times 16$ grid map. Each cell on the map is associated with a 27-dimensional feature vector that encodes the type of unit (None, Resource, Base, Barrack, Worker, Light, Heavy, Ranged) with different capabilities, its owner (Team Blue (ally), Team Red (enemy), None), number of resources it carries $(0,1,2,3,\geq 4)$, health value of the unit, termed *hit points* (HP) $(0,1,2,3,\geq 4)$, and current action of the unit (none, move, harvest, return, produce, attack), giving rise to a $16 \times 16 \times 27$ tensor as the observation state of one frame. Each action assigned to a unit on the board is represented by an 8-dimensional vector that encodes the source unit's location, the action type, and action-specific parameters.

During game play, the RL agent (or the human player) takes up the role of player 1 (ally), while the opposing entity, player 2 (enemy), is represented by the COAC enemy bot, the overall champion in the 2020 MicroRTS AI Competition (Ontanon, 2013).

## B. Human Expert Study Details

### B.1. Behavior Prediction Examples

Figure 6 presents two examples used in the behavior prediction tasks administered to participants. Participants in the control group had access only to the current game state (see Figure 6(a)) and their previously observed states. In contrast, participants in the experimental group could not navigate to prior states but were provided with local causal models, as shown in Figure 6(b). Partial outcomes, indicating the ground truth, are illustrated in Figure 6(c).

Example (1) was presented with the following question: *Given the board state below (1 ally base, 8 ally workers, 1 enemy base, 1 enemy barrack, and 2 enemy workers), what would be the strategic plan of the agent (ally) to counteract the enemy's action of building heavys?*

Participants were asked to select one of the following options:

- Harvest Resource(s) $\rightarrow$ Build more Worker(s).

- Build a Barrack $\rightarrow$ Build more Worker(s).

- Build a Barrack $\rightarrow$ Build more Ranged(s).

- Build a Barrack $\rightarrow$ Build more Light(s).

- Build a Barrack $\rightarrow$ Build more Heavy(s).

- Build a Barrack $\rightarrow$ Build more Ranged(s) and Build more Light(s).

- Build a Barrack $\rightarrow$ Build more Ranged(s) and Build more Heavy(s).

- Build a Barrack $\rightarrow$ Build more Light(s) and Build more Heavy(s).

Example (2) was presented with the following question:

*Given the board state below (1 ally base, 1 ally barrack, 8 ally workers, 2 ally lights, 1 ally heavy, 1 ally ranged, 1 enemy barrack, 1 enemy heavy), which type of ally unit would be affected first if the enemy increased the number of heavys?*

The answer choices were: Heavy, Light, Ranged, Worker, Base, Barrack

### B.2. Assessing Explainability and Trust

Explainability, in particular, was assessed through participants' agreement with the following claims:

- **Understand**: *The explanations helped you understand what the agents were doing.*

- **Satisfying**: *You were satisfied with the explanations provided to understand the agents' strategies.*

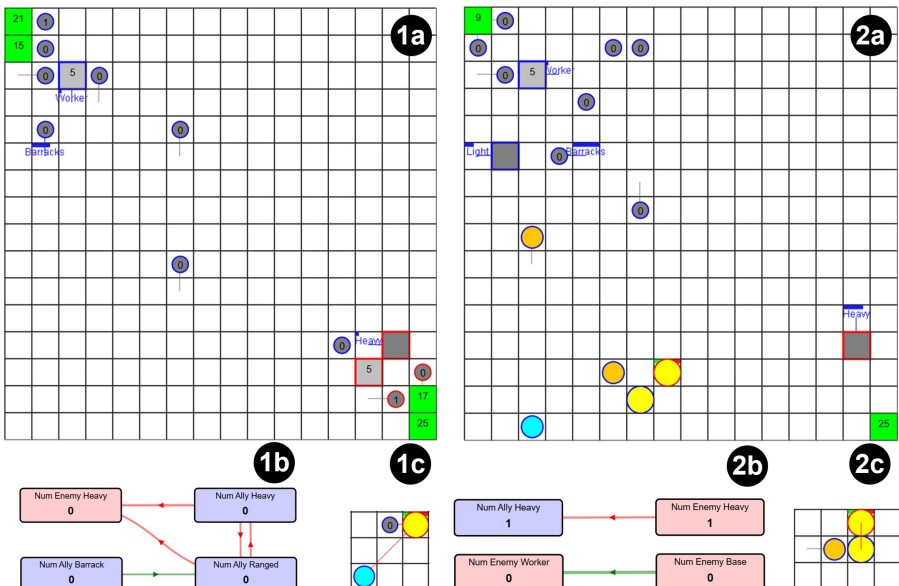

*Figure 6.* Two examples (1) and (2) of game states. (a) Example game states sampled from the task. (b) Local causal models shown to the participants (c) Ground truth shown as partial outcomes.

- **Sufficient**: *The explanations included enough information for you to understand the agents' behavior.*

The global causal graph obtained an average rating of 4.50 for *understand*, 4.00 for *satisfying* and 3.75 for *sufficient*, indicating an overall positive impression. However, despite being plausible, the absence of visible edge influences impaired the explainability of the global causal graphs.As E5 noted: "It makes sense that the effects of edges are not shown in the global causal graphs, as aggregated effects can be misleading when feature correlations are not monolithic. Still, users might want to have an overall impression of directionality." The local causal models received average ratings of 4.63 for *understand*, 4.50 for *satisfying*, and 4.38 for *sufficient*, indicating that participants found the explanations to be highly effective.

Trust was assessed using the following statements:

- **Confident**: *I am confident that the agents will act in accordance with the explanations provided.*

- **Predictable**: *Based on the explanations provided, I feel I can anticipate the agents' next moves.*

- **Reliable**: *The explanations provided are consistent with the agents' strategies.*

Participants rated the local causal models an average of 4.38 for *confident*, 4.00 for *predictable*, and 4.75 for *reliable*, suggesting a strong level of trust in the local models' ability to support explanatory tasks.

### B.3. User Interface

Figure 7 presents the user interface employed during the exploratory expert study.

## C. Collective Expert Elicitation for Representation Extraction

CausalXRL relies on a representation map $\Phi$ that converts each raw RL experience $e^{(t)} = (s^{(t)}, a^{(t)}, r^{(t)}, s^{(t+1)})$ into a vector of interpretable representations $\kappa^{(t)} = \Phi(e^{(t)})$. In simple benchmark domains, this map can often be specified directly from the state variables. In richer multi-agent environments, however, a single researcher may overlook strategic abstractions that are obvious to domain experts, or may define features at a granularity that is too low-level to support useful explanations. We therefore frame representation extraction as a collective expert elicitation problem: experts jointly construct a shared

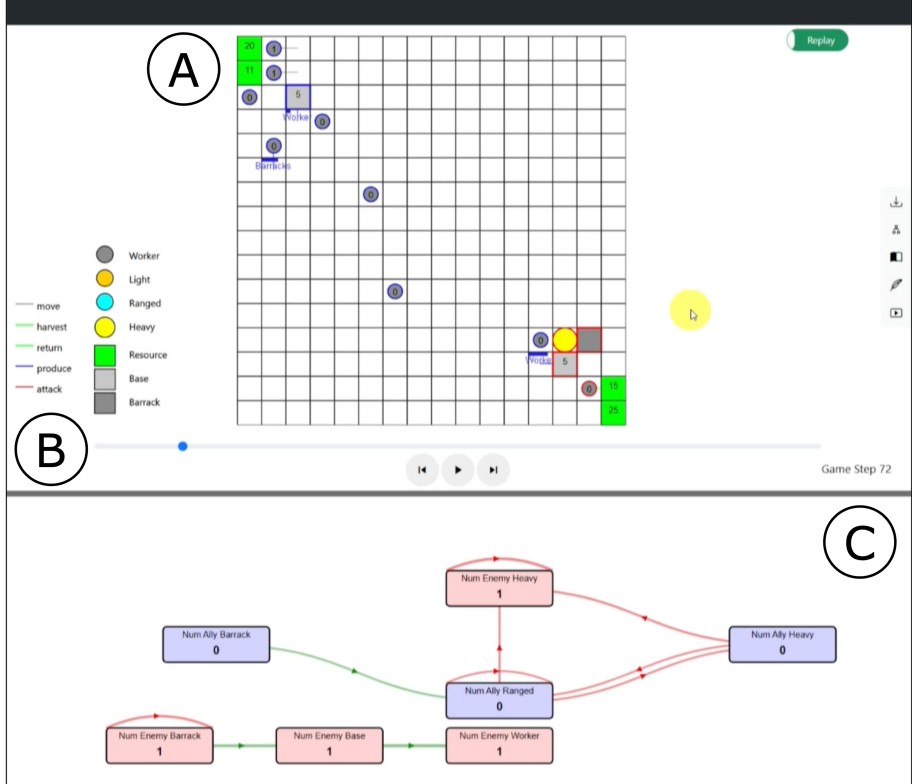

*Figure 7.* A screenshot of CausalXRL's user interface for MicroRTS. It includes (A) the interactive game board where users can visualize game states and modify current one through clicking events, (B) the slider to navigate through episode's game step, and (C) the local causal model corresponding with the current game state.

vocabulary of observable, causally meaningful, and human-interpretable representations before CausalXRL learns causal dependencies among them.

Our elicitation protocol follows a wisdom-of-the-crowds design inspired by multi-turn contextual argumentation systems such as Cicero (Chen et al., 2019). The key idea is that expert agreement is more reliable when participants first make independent judgments and then revise them after seeing concrete opposing arguments grounded in the same context. In our setting, the "crowd" is not responsible for directly labeling the causal graph. Rather, it is used to construct the representation codebook that specifies the candidate variables to be included in the ensuing causal graph construction.

**Stage 1: Independent proposal.**    Experts first inspect a stratified set of replay clips, states, and trajectories sampled from the trained agent. The sample is chosen to cover common play, failure cases, rare events, and transitions near large reward changes. Each expert independently proposes candidate representations. For every candidate, the expert records: (i) a short name, (ii) a natural-language definition, (iii) the intended value type, and (iv) a rationale for why the representation is expected to be useful for causal explanation.

**Stage 2: Contextual argumentation.**    The proposed candidates are then merged into discussion items. Synonymous candidates are grouped, while conflicting candidates are surfaced explicitly. Conflicts usually concern granularity, observability, or causal usefulness. For each conflict, experts are shown the same local context: the relevant replay segment, current candidate extractor output, and the rationales submitted by other experts. Experts then provide short arguments and counterarguments, focusing on whether the candidate is understandable, observable from the environment, and likely to capture a strategic mechanism rather than a superficial correlation. This multi-turn discussion prevents the representation set from being determined by an unexamined majority vote: a minority expert can still change the final codebook by pointing to a context in which a proposed feature fails or by identifying a missing abstraction.

**Stage 3: Consensus aggregation.** After discussion, experts cast final accept, reject, or merge judgments for each candidate. We retain a representation if it satisfies four criteria: (1) it is observable or extractable from the replay data, (2) its definition is semantically clear to multiple experts, (3) it is plausibly relevant to behavior or outcome changes, and (4) it is not redundant with a simpler accepted representation. When confidence scores are collected, the consensus score for candidate $j$ can be computed as

$$S_j = \frac{\sum_{i=1}^m c_{ij} v_{ij}}{\sum_{i=1}^m c_{ij}},$$

where $v_{ij} \in \{0, 1\}$ denotes expert $i$'s final accept/reject vote and $c_{ij}$ denotes that expert's confidence. Candidates above a preset threshold are accepted; borderline candidates are either refined and re-evaluated or kept as optional diagnostic variables.

**Stage 4: Operationalization and validation.** The accepted representations form a consensus codebook $\mathcal{B} = \{b_1, \ldots, b_n\}$. The extraction rules are implemented as deterministic or learned feature extractors $\phi_j$, yielding $\kappa^{(t)} = (\phi_1(e^{(t)}), \ldots, \phi_n(e^{(t)}))$. Before causal learning, we validate these extractors against held-out expert annotations on unseen states or clips. This validation checks whether the implemented extractor matches the collective expert meaning of the representation.

The final output of the process is a set of interpretable representations for the environment. In the MicroRTS setting, such sets can include high-level strategic features such as counts of ally and enemy unit types, base and barracks status, resource pressure, attack pressure, and production capacity. CausalXRL then learns dependencies among these agreed-upon representations from replay data. This separation is important: experts define the language in which explanations are expressed, while the causal inference module estimates which dependencies are supported by the agent's observed behavior. As a result, the learned global and local causal models remain data-driven while using variables that are meaningful to human analysts.

## D. Sensitivity Analysis

We conduct a sensitivity analysis to examine the robustness of the learned causal dependencies under different percentile thresholds applied to the BART posterior samples. Specifically, we construct global causal graphs using central posterior intervals of 90%, 95%, and 99%. As shown in Figure 8, the resulting graphs allow us to assess whether the inferred causal structure remains stable as the confidence threshold becomes more stringent. Dependencies that persist across all three thresholds indicate robust causal relationships, whereas edges that appear only under less stringent thresholds should be interpreted as weaker or less stable.

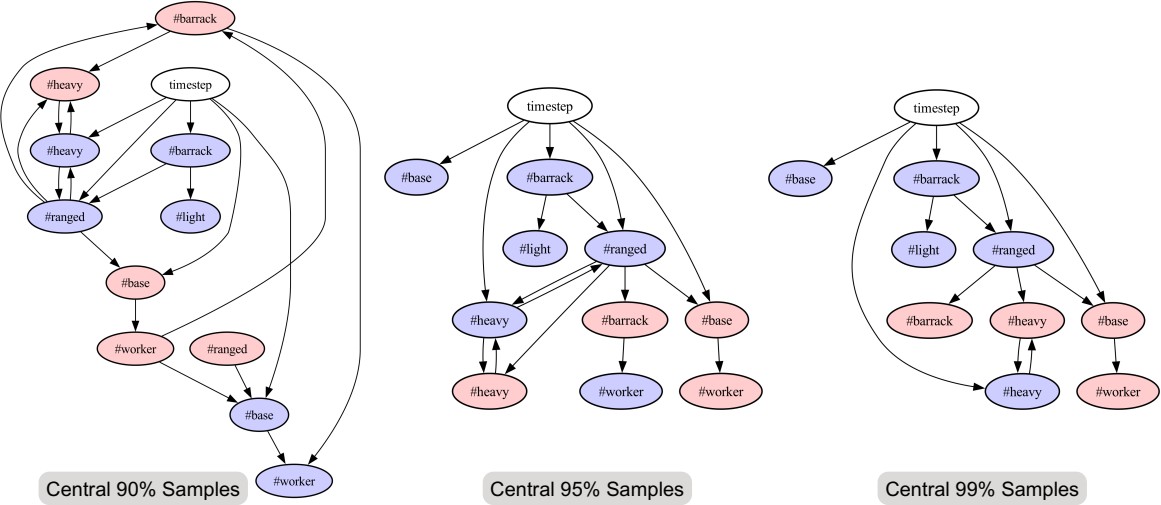

*Figure 8.* Global causal graphs under different CATE posterior thresholds. From left to right, the threshold increases, corresponding to a more conservative criterion for edge inclusion.

In the presented example, increasing the central posterior interval from 90% to 95% removes five edges: *Blue Base→Blue*

*Worker*, *Red Worker→Blue Base*, *Red Barrack→Red Heavy*, *Red Worker→Red Barrack*, and *Red Ranged→Blue Base*.

Further increasing the interval from 95% to 99% removes three additional edges: *Red Barrack→Blue Worker*, *Blue Heavy→Blue Ranged*, and *Blue Ranged→Blue Heavy*.

This pattern suggests that these removed edges are more sensitive to the percentile threshold, whereas edges that persist across the 90%, 95%, and 99% settings provide stronger evidence of robust causal dependencies.

