# OpenReview forum: "CausalXRL: Explainable Reinforcement Learning through Causal Graph Reasoning"
_ICML.cc/2026/Conference — ICML 2026 regular_

### Official Review · Reviewer_ZTUo · 2026-03-12

**Soundness:** 3
**Presentation:** 3
**Significance:** 2
**Originality:** 3
**Overall Recommendation:** 4
**Confidence:** 3

**Summary:**

This study claims to discuss the concept of CausalXRL, a novel framework that utilizes Rubin Causal Models and Bayesian Additive Regression Trees to extract human-interpretable representations and infer causal dependencies without requiring predefined structures or computationally expensive independence tests. By generating Global Causal Graphs for overarching policy insights and Local Causal Models for situational counterfactual reasoning, the framework provides a multi-level explanatory structure that outperforms traditional post-hoc methods like SHAP in diagnostic fidelity. Validation through benchmarks and a qualitative expert study in MicroRTS demonstrates that CausalXRL significantly improves human understanding and strategic predictability.

**Compliance With Llm Reviewing Policy:**

Affirmed.

**Final Justification:**

I apologize for the delay. Thanks the authors for the comment, but i will maintain my score due to the not enough evidence on the benefit of the method in the not valuable experiment.

**Key Questions For Authors:**

- What happens if the "expert" gives you bad features?
- How do we know the agent is actually thinking this way?
- With a sample size of N=8 experts, how did you control for 'confirmation bias,' where experts might rate an explanation as 'satisfying' simply because it aligns with their existing game knowledge, rather than accurately reflecting the agent’s unique logic?

**Limitations:**

### The "Expert-in-the-Loop" Bottleneck
The whole framework depends on "expert-specified interpretable representations." If the expert misses a key feature, the causal graph will be wrong or incomplete (omitted variable bias). This limits how "autonomous" the explanation generation actually is.

### Stationarity Assumptions
RL environments are often non-stationary (the optimal strategy changes over time). The paper doesn't clearly explain how the Global Causal Graph handles a policy that evolves or shifts radically midway through training.

### Simulated vs. Real Complexity
While MicroRTS is a step up from CartPole, it's still a grid-based, low-complexity environment. It’s unclear if the BART-based inference would collapse under the weight of hundreds of variables in a modern AAA game or a real-world robotics task.

**Strengths And Weaknesses:**

## Strengths
### High Scalability
CausalXRL uses a structure learning algorithm with $O(n^2)$ complexity. This makes it viable for complex, high-dimensional environments like RTS games.
### Dual-Level Explanations
The combination of Global Causal Graphs and Local Causal Models provides a comprehensive toolkit for both developers debugging a system and end-users trying to understand specific moves.

## Weaknesses
### The "Expert Echo Chamber"
Because the validation relies on expert intuition, there’s a risk that the model is just telling the experts what they want to hear. If the agent makes a weird but effective move that doesn't fit a "human" strategy, the experts might call the explanation wrong even if it’s technically what the AI was thinking.
### Sensitivity to Time-Step Confounding
As noted in the study, if "time" isn't included as a covariate, the model can learn spurious correlations based on game stages. This suggests the model may struggle in environments with high temporal non-stationarity where causal relationships shift over time.
### Inflated Trust Scores
It’s easy to get a 4.75/5 rating when you have a small group of people who are essentially being "guided" through the tool. Without a large-scale, blind test, those high trust metrics feel more like a pat on the back than a rigorous scientific proof.

### Subjectivity in Global Models
Expert feedback indicated that global graphs, while accurate, can be less intuitive because they lack the specific directionality and influence weights found in local models, potentially leading to ambiguity for non-expert users

---

> ### Author Rebuttal · Authors · 2026-03-30
>
> Thank you for the constructive feedback. In the revision, we will sharpen the paper’s scope by explicitly stating the dependence on representation quality, clarifying that our method learns a faithful causal surrogate rather than the agent’s literal internal reasoning, and framing the expert study as controlled but exploratory.
>
> **Dependence on expert-specified features**
>
> We agree this is a limitation. CausalXRL operates over human-interpretable representations, so omission of key features can lead to incomplete causal graphs. We will state this more explicitly in the limitation. At the same time, we will also include details on the collaborative guideline used during feature extraction step to mitigate this issue, which is a 'wisdom of the crowds' benchmark similar to Cicero (Chen et al., 2019).
>
> **Faithfulness vs. literal internal reasoning**
>
> Our claim is not that we recover the agent’s literal internal reasoning. Rather, we learn a human-interpretable causal surrogate intended to be faithful to the trained policy’s behavior. This is why the computational evaluation focuses on behavior prediction: the learned local causal models are tested against the agent’s actual actions across multiple environments. We will revise the wording to make this scope precise.
>
> **Small expert study and confirmation bias**
>
> We agree that the human study should be interpreted as qualitative and exploratory, not as a statistically powered trust study, and we will make this framing more prominent. To distinguish merely satisfying explanations from explanations that better reflect agent logic, we relied on performance in the controlled behavior-prediction task. A procedural detail omitted from the current manuscript is that participants in both groups first predicted behavior before viewing the assigned XAI tools. This captures prior knowledge separately from the improvement attributable to the tool, which helps mitigate confirmation bias.
>
> **Time-step confounding and non-stationarity**
>
> We agree that local causal models may be challenged in environments with strong temporal non-stationarity. To mitigate this, we will make the design guidance explicit: when stage-of-episode effects may confound inference, game step or time step should be included in the representation set. We will also strengthen the limitations discussion accordingly.
>
> **Scalability beyond MicroRTS**
>
> We agree that the current empirical evidence supports benchmark environments and MicroRTS, not yet AAA-scale games or robotics-scale deployments. Our scalability claim is narrower: relative to CIT-based approaches, our structure-learning procedure has better worst-case complexity and is intended to remain practical with dozens of interpretable variables. We will revise the text to avoid overclaiming beyond the evaluated regime.

---

### Official Review · Reviewer_FY6i · 2026-03-12

**Soundness:** 2
**Presentation:** 2
**Significance:** 3
**Originality:** 3
**Overall Recommendation:** 4
**Confidence:** 3

**Summary:**

The paper focuses on explainable reinforcement learning to make the agents' decision-making transparent and interpretable. The authors introduce CausalXRL, which builds representation pools and creates a global causal graph across all episodes and a local causal graph for context-aware reasoning. The framework is evaluated through both quantitative analysis of agent performance and explanation fidelity in benchmark RL environments and a qualitative expert study assessing the RTS game.

**Compliance With Llm Reviewing Policy:**

Affirmed.

**Final Justification:**

I would like to keep my positive score.

**Key Questions For Authors:**

It would be beneficial to further discuss how the learned causal graphs can be integrated into or support the reinforcement learning process itself. Can the learned causal graphs also be used to improve the reinforcement learning process itself (e.g., policy optimization, exploration)? If so, how would this integration work?

Since the framework relies on expert-specified interpretable representations, how are these representations practically defined and extracted in complex environments? Does the method require domain experts for each new task? How robust is this method to the quality of such predefined representations? Also, how is the scalability when involving more agents, more complex tasks?

Could the author provide cases to explain what a learned causal graph looks like? Especially the difference between the global causal graph and the local ones.

**Limitations:**

Yes.

**Strengths And Weaknesses:**

I appreciate the idea of building higher-level causal graphs (beyond factorized state and action representations) to explain agent behavior. The paper is generally well written and easy to follow.

However, the paper could benefit from more illustrative examples, and some figures could be made more self-contained. For instance, it is unclear what SLA refers to.

I am also concerned about the generalizability and robustness of the proposed method. The approach appears to rely heavily on the quality of expert-specified interpretable representations. If the representations change, the resulting causal graph may also change significantly. Moreover, the paper lacks sufficient details on how these expert-specified representations are extracted. In particular, the definition and implementation of the function $\Phi$ are not clearly explained. It would be helpful if the authors clarified the scope and assumptions of this component.

---

> ### Author Rebuttal · Authors · 2026-03-31
>
> We thank the reviewer for the constructive comments. In the revision, we will improve figure clarity, define notation and abbreviations more carefully, expand the description of $\Phi$ and representation extraction, add clearer examples contrasting global and local causal models, and sharpen the discussion of limitations and future integration with RL.
>
> **Figure clarity and missing definitions**
>
> We agree that the figures can be made more self-contained. We will explicitly define all abbreviations, including “SLA” as structure learning algorithm. In our framework, this refers to the BART-based procedures for global causal graph construction and local causal model learning.
>
> **Scope of the method and $\Phi$**
>
> We will clarify that CausalXRL does not address automatic concept discovery. Its main contribution is a causal explanation framework that operates once a human-interpretable representation space is available. We will also better define $\Phi$, which maps raw experience to interpretable representations. In MicroRTS, $\Phi$ is implemented through deterministic mappings from raw state to high-level strategic features (It has a closed form function, due to the environment. We will add more details about it in the appendix); when such mappings are not directly available. For example, when the raw state is an image, a concept bottleneck-style pipeline that collect data of the form (x,c,y) is needed. Here, x is the input, c is the concept vector, and y is the label.
>
> **Dependence on expert-specified features**
>
> We agree this is a limitation. CausalXRL operates over human-interpretable representations, so omission of key features can lead to incomplete causal graphs. We will state this more explicitly in the limitation. At the same time, we will also include details on the collaborative guideline used during feature extraction step to mitigate this issue, which is a 'wisdom of the crowds' benchmark similar to Cicero (Chen et al., 2019).
>
> **Generalizability and scalability**
>
> We will clarify that generalizability depends on whether a task admits a meaningful interpretable representation interface. On scalability, our graph construction scales as O(n^2), which is substantially more efficient than CIT-based alternatives, though very large causal networks may still be challenging.
>
> **Integration with RL training**
>
>  We agree this is promising future work. The current paper focuses on post-hoc explanation, but the learned graphs could potentially support exploration, debugging, and human-in-the-loop intervention.
>
> **Global vs. local causal models and their examples**
>
> We will make this distinction more explicit: global graphs summarize stable cross-episode causal structure, while local models are state-specific, sparse, and attach signed influence strengths to edges for context-aware reasoning and counterfactual analysis.
>
> Aside from current examples provided now ( Figure 2(a) for a global causal graph and Figure 2(c) for a local causal model ) we will include more examples of them in the appendix across different RL environments.

---

> > ### Author Rebuttal · Reviewer_FY6i · 2026-04-01
> >
> > Thanks for the authors' effort in addressing my concern. After reading the response, I decide to keep my scoring.

---

### Official Review · Reviewer_YUoj · 2026-03-19

**Soundness:** 2
**Presentation:** 3
**Significance:** 3
**Originality:** 3
**Overall Recommendation:** 2
**Confidence:** 3

**Summary:**

This paper proposes CausalXRL, a framework for explaining reinforcement learning agents through causal reasoning. The method consists of three stages: representation extraction, causal dependency inference, and explainable causal models. It extracts human-interpretable representations from trajectories, treats them as observational data, and uses Rubin causal models with BART to estimate conditional average treatment effects (CATE) between representations. Based on these estimates, the paper constructs both a global causal graph and local causal graph to explain agent behavior at the global and situational levels. The paper reports behavior prediction and structural learning efficiency on four RL benchmarks, and includes a user study with 8 RTS experts in MicroRTS.

**Compliance With Llm Reviewing Policy:**

Affirmed.

**Key Questions For Authors:**

1. The quantitative evaluation mainly measures action prediction and graph-construction efficiency. Could the authors clarify why these proxies are sufficient to support claims about explanation fidelity or the validity of the learned causal structure?

2. How sensitive are the global and local graphs to the 2.5% / 97.5% percentile thresholds used in Algorithms 1 and 2? If small threshold changes materially alter the graph structure, that would affect the robustness of the method. A sensitivity analysis would be very helpful.

3. Could the authors either compare against the closest XRL / causal explanation baselines, or explain more concretely why direct comparison is not feasible in the chosen environments?

**Limitations:**

Yes.

**Strengths And Weaknesses:**

Strengths:

1. The method proposed in this paper seems reasonable. It combines a global causal graph and local causal graph, the former provides a structured high-level overview, while the latter supports context-specific analysis and counterfactual reasoning. The causal perspective for XRL is well motivated.

2. This work has a clear motivation for avoiding conditional-independence-test-based structure learning and instead using BART-based causal dependency inference for better scalability.

Weakness:

1. The representation layer depends on expert-specified concepts, which is understandable from an interpretability perspective, but it also makes the no-unobserved-confounding assumption harder to assess in complex settings. If relevant strategic variables are not covered by the representation layer, the causal interpretation of the learned edges becomes less clear.

2. It is unclear to what extent the method captures delayed or long-horizon causal effects, and this aspect is not sufficiently validated in the paper.

3. The baselines are mainly the PC algorithm, linear models, and decision trees, leaving the comparison against the closest XRL / causal explanation methods somewhat limited.

4. The user study is informative, but the conclusions should remain more cautious. The paper explicitly frames the results as suggestive and exploratory evidence; with 8 experts and 4 MCQ scenarios, the study supports the claim that the explanation format is promising, but it is not yet strong evidence for broader trust or usability claims.

---

> ### Author Rebuttal · Authors · 2026-03-31
>
> We thank the reviewer for the careful reading and constructive feedback. Below we address the main concerns and clarify how we will revise the paper.
>
> **Representation Assumptions and Causal Sufficiency**
>
> In complex environments, omitted strategic variables can weaken the no-unobserved-confounding assumption. We will revise the paper and state this more explicitly in the limitation. At the same time, we will also include details on the collaborative guideline used during feature extraction step to mitigate this issue, which is a 'wisdom of the crowds' benchmark similar to Cicero (Chen et al., 2019).
>
> **Delayed and Long-Horizon Effects**
>
> Our view is that CausalXRL captures causal effects at the level of strategic state changes reflected in the interpretable representations. If a change in the raw state does not alter the extracted high-level features, then it does not change the explanatory abstraction we aim to model. For example, a unit moving or standing may matter tactically, but if it does not change the coarse strategic features, it provides little information gain at the explanation level considered here. In that sense, what counts as a “long-horizon effect” depends on how much temporal and strategic dynamism is preserved in the representation layer. We will clarify this framing and state more explicitly that the method models causal dynamics at the abstraction level of the extracted representations.
>
> **Why We Use Action Prediction to Evaluate Fidelity**
>
> The rationale of using action prediction as a proxy to evaluate fidelity is that direct evaluation of arbitrary local edge dependencies is difficult because node frequencies differ across local graphs, and edge strengths are continuous quantities whose evaluation becomes a hard-to-interpret regression problem. We therefore constrain the target node to be an action node, which turns the evaluation into a classification problem and provides an interpretable proxy for whether the upstream causal structure captures behaviorally relevant dependencies.
>
> **Sensitivity to the 2.5% / 97.5% percentile Thresholds**
>
> This is a very helpful suggestion. The current threshold follows the standard 95% credible-interval rule of thumb for BART posterior samples, and was chosen as a conservative decision rule to reduce false positives. That said, we agree robustness should be demonstrated directly. In the revision, we will add a sensitivity analysis showing how graph structure and downstream results change under different threshold choices.
>
> **Choice of Baselines**
> We agree the baseline section can be strengthened. For causal discovery, PC is one of the most widely used and strongest standard baselines, which is why we used it for the main efficiency comparison. In the revision, we can expand the comparison to additional causal discovery methods such as GES and FCI to better illustrate the trade-offs between discovery quality and runtime. For behavior prediction, we chose linear models and decision trees because they are common, interpretable surrogate baselines in XRL; we can also add a stronger tree-based baseline such as random forest while still maintaining comparability to explainable surrogate approaches.
>
> **Comparison to Closest Causal-XRL Methods**
> We agree we should explain the lack of direct comparison more concretely. The key difference is that in AIM-style methods each edge is tied to a single action, and the edge strength explains the contribution of that action to a state transition. In our framework, edges are not action-specific; they capture aggregated state-transition dynamics that may reflect multiple actions and interactions from both agents. This distinction is especially important in MicroRTS, where we intentionally abstract away fine-grained tactical details and instead explain strategic state evolution. Under this setting, AIM would predict the most influential action, whereas our framework explains how the current strategic state influences the next strategic state. We will make this mismatch in explanatory target and evaluation protocol more explicit in the revision.
>
> **Interpreting the User Study**
> We agree and will make the wording more cautious. The paper already describes the study as providing suggestive and exploratory evidence, and we will further emphasize that with 8 experts and 4 MCQ scenarios, the results support the promise of the explanation format in MicroRTS but should not be taken as broad evidence for trust or usability at larger scale. We will revise the claims accordingly.

---

> > ### Author Rebuttal · Reviewer_YUoj · 2026-04-08
> >
> > Thanks for the reply. The authors have addressed a large portion of my concerns; however, I remain unsatisfied with the treatment of baselines, particularly with respect to causal discovery methods and related Causal-XRL approaches. To some extent, more comprehensive comparisons and experiments with state-of-the-art methods are still needed in the current version. Otherwise, the evidence provided is insufficient to support publication at this stage.
> >
> > I therefore maintain my evaluation.

---

### Official Review · Reviewer_Xe6L · 2026-03-20

**Soundness:** 3
**Presentation:** 2
**Significance:** 3
**Originality:** 2
**Overall Recommendation:** 4
**Confidence:** 4

**Summary:**

The paper introduces an approach to explainability that leverages causality for XRL purposes. Broadly, the proposed approach works by extracting a causal model from offline experience data from a trained RL policy, then presenting that causal model to the user. The paper also provides a qualitative study to evaluate the prototype.

**Compliance With Llm Reviewing Policy:**

Affirmed.

**Final Justification:**

I argue to be on the ACCEPT side of accept/reject because the paper seems to make a solid contribution in presenting causal graphs to explain policies in MicroRTS. The rebuttal answers were reasonably convincing, and I do think the next revision of this paper will be strong. However, my main concern (and why I only argue for WEAK accept) is that the presentation of the original submission suffers from some clarity deficiencies that may remain in a camera ready if ICML accepts this paper. This means another round of review may be warranted.

**Key Questions For Authors:**

When consulting Figure 2, I had several questions and comments:
1. The polarity is unclear. By this I mean that the arrow does not indicate if increases to "red #barrack" cause increases or decreases in "blue #worker". Based on oppositional relationship one might assume a decrease, but the visualization should indicate this information and make some recommendations for how to improve the visualization based on participant feedback. The lone participant quote flicks at this, but the paper makes no recommendations in response to the feedback. (Sidenote, L305 notes that certain edges have greater strength, but this is also not visible in Figure 2).
2. The graph feels sparser than it should be with respect to edges. For example, there is no edge between "blue #light" and "red #heavy". Presumably, heavies counter lights, so while this is an unfavorable matchup, lights presumably CAN kill heavies, meaning I would expect this edge to be present. There are other edges that follow a similar argument.
3. The graph feels sparser than it should be with respect to nodes. As an example, "red #barrack" does not directly interact with "blue #worker", therefore this edge is somewhat ill-defined. My assumption is that there is a missing node: their barracks will produce some sort of units, which will then kill my workers. Adding the node causes each edge to be well defined.
4. How does this structure handle multi-agent behavior? S4.2 claims CausalXRL's "explanatory value in complex, multi-agent settings"? Since each unit can be attacking a different target, how is that visible in the explanation, given that it appears there is only 1 outbound edge to an enemy from "#heavy" in the global causal model.

Other questions and comments:

5. Why is "accurate behavior prediction" the proper metric for S4.2? The paper has not argued that these policies exhibit class balance. Therefore, I would expect this analysis to examine F1
6. Why split the explanation and trust metrics across the local and global causal models? To me, all metrics seem well defined for all models.

**Limitations:**

The boilerplate impact statement is present, which seems sufficient to me.

**Strengths And Weaknesses:**

Strength - XRL and causality are both important lines of work, marrying the two is worthy

Strength - Conducting a user study to evaluate explainability is laudable

Weakness - Paper is hard to follow in several places

Weakness - Qualitative study has an odd presentation that mismatches claims and evidence

Overall, I think this paper needs a round of revision, but that the future version should be very solid, provided there aren't deep flaws in the underspecified elements.


Specific Issues
---------------

 - Paper is hard to follow in several places

1. Introduction L72-77 is the first instance. The main problem here is that the meaning of this edge is ill-specified. From context much later in the paper, it seems like the idea is that when one faction has attacking units the other faction will have a reduction in the number of bases? This is confusing for two reasons. First, often attacking units attack other units and do not impact the base at all. Second, attacking units have the capability to retreat. Last, as written, the paper seems to diminish how much insight folks can gather by "simply watching the game". A number of XAI works have found the "What" Lim-Dey intelligibility type to be very powerful, which is exactly that experience - watching the game.
2. The next is in S3.3.1, with the directed edge "blue barrack -> blue light". At this point, the reader has not seen what a "Light" is only "Ranged", which is a little more self-explanatory. Later the paper introduces Medium and Heavy, so it seems these are some kind of mobile troop, presumably melee. Given that the paper relies so heavily on MicroRTS, a brief primer early in the paper would be beneficial. Alternatively, the few examples that rely on domain knowledge could add increased context.
3. The most important passage that was unclear was in S3.3.2, from L307-L318. Again, the causal chain in the first part is hard to interpret, partially because the arrows have different meanings for the same symbol (I think the first one is "produce" and the second is "attack"). More importantly though, it seems like the evidence in the passage about "strategy validated" is a proof by example. Last, the point about "blue heavy -> red heavy" seems to be true about any attacking unit, in that it is possible for them to get a unit kill.
4. Figure 2 is far too small. This is the most important figure in the paper, it needs more real estate. The rule of thumb for figure font is that it should be similar in size as the body text, this is about half that size.
5. The second paragraph of S5.2 does not make sense. It sounds as if the participants' task was to construct a list of edges found in the graph, but that does not seem like an accurate characterization due to being far too simple to be useful as a task.

- Qualitative study has an odd presentation that mismatches claims and evidence

On L30 abstract claims "Results show CausalXRL enhances human understanding...". This claim is improper since it relies on a pre-then-post comparison and qualitative studies are focused on phenomenon elicitation, not quantitative comparison (as the paper also states astutely in the start of S5.3). The conclusion then echoes the improper claim. Additionally, the paper does not indicate if the protocol went through an IRB (or similar) approval process, whether participants gave consent, or what compensation participants received, if any. The protocol does not specify what happens when a participant fails a quiz in S5.2. Later in that section, we learn that the study employed a crossover design, but the analysis does not utilize this fact in any way. The paper spends too much time reporting quantitative data and should contain more quotes and synthesis of the feedback to improve the human-facing elements. This is because, as the paper states, small qualitative study designs are not appropriate for making comparative claims about the sample, which the paper also does. I suggest consulting CHI or IUI for papers that report on user studies of this kind for inspiration. Both of those communities also have literature this paper should cite.


Minor Issues
------------

- The paper contains a large number of "runt" lines. Removing them will free up valuable space for larger figures, more details, etc.

- Introduction L61 has weird period location and spacing around the Yu citation.

- S3.1 L196, suggest adding a clause mentioning what the Phi is (seems to be the mapping to latent representation)

- S3.2 L206 there is a missing space in the last sentence of the paragraph following Eq 1

- language is a little strange in places:

1. S3.3 L241 "similar to reverse engineer RL agent policies" -> reverse engineering?
2. S3.3.1 L273 "treatment effect which unveiling" -> which unveils?
3. S3.3.2 L269 "can assist analyst to determine" -> assist analysts?
4. Same sentence... "in what direction the tweaking is needed to approach" -> in what direction they should adjust to approach?

- Body text refers to Linear models (LM), but Table 1 refers to LR. I think this should read LM

- S4.2 claims "an advantage that should grow in larger environments". There is an old adage in film that applies to research "show, don't tell."

- missing commas around the clause "especially novice players" in S5.1

- Missing space before the quote of E5. Also italicize change in speaker. Also the language E5 uses in this quote seems pretty advanced for a random RTS player...

- references are untidy. Check for capitalization of proper nouns and acronyms (e.g., Atari, XAI), etc.

---

> ### Author Rebuttal · Authors · 2026-03-31
>
> Thank you for the careful review and insightful questions. We appreciate the concrete suggestions for improving the paper’s clarity, we will revise accordingly.
>
> **Clarity of the causal edges and MicroRTS examples**
>
> We agree that several causal edge examples currently assume too much domain knowledge and do not explain edge semantics carefully enough. In the revision, we will add a short MicroRTS primer, revise the examples in S3.3 to state the meaning of edges more consistently, and enlarge Fig. 2 for readability. We also agree that Fig. 2 currently under-specifies polarity and edge strength. We will therefore add a legend and clearer description so that edge direction, polarity, and relative strength are visually and verbally explicit.
>
> **Qualitative studies results**
>
> We agree that the qualitative study was intended as exploratory and should not have been framed with strong quantitative claims. We will revise to present the findings as suggestive evidence that participants found the explanations useful, rather than as a statistically powered comparison. In addition, we will strengthen the qualitative reporting by adding more participant quotes, synthesizing the feedback more clearly, and including the missing study-protocol details such as IRB information and compensation. We will also better align the framing of the study with prior HCI work, including Buçinca et al. (2020), to appropriately interpret the findings.
>
> **Why some seemingly plausible edges are absent**
>
> The absence of an edge in our graphs should not be interpreted as “this interaction is impossible in the game.” Rather, it means that under the current representation set, the behavior of the trained agent, and our conservative edge criterion, we did not identify a stable direct causal dependency. This is especially relevant for examples such as “blue Light → red Heavy”: while such an interaction is mechanically possible in MicroRTS, the learned graph is intended to reflect the policy behavior present in the replay data, not the full capability graph of the game. In the episodes used here, the ally policy tends to prefer Heavy production and associated attack patterns, so dependencies involving Light units may not survive conservative structure learning. We will make this distinction explicit in the revision.
>
> **Why some edges appear without an explicit mediator node**
>
> This is a good observation. The global graph summarizes aggregate direct effects across episodes, so when a relation is mediated through multiple alternative units or pathways, those finer-grained mechanisms may not appear as a clean chain in the global view. For example, a dependence between enemy Barracks and ally Workers may be mediated through multiple downstream unit types; under conservative averaging, those heterogeneous mediated effects can be diluted or “washed out” at the global level. Our local causal models are better suited to exposing these context-dependent pathways because they retain state-specific edge strengths.
>
> **Handling multi-agent behavior**
>
> Our claim about multi-agent explanatory value is at the level of aggregate strategic behavior, not per-unit target assignment. The representations in MicroRTS are high-level counts of unit/building types, and the learned causal models explain dependencies among those team-level strategic variables. The global causal graph in Fig. 2 is one learned example and should not be read as implying that every trained agent will have only one outbound cross-team dependency. In addition, because our default global graph learner is intentionally conservative, some dependencies may be suppressed by averaging heterogeneous effects. We will revise the text to include additional examples comparing more conservative and more exploratory graphs and different trained agents.
>
>
> **Why use behavior-prediction accuracy in S4.2**
>
> Our intent in S4.1 was to treat the local causal model as a surrogate explanation model and assess whether it can recover the trained agent’s next action from the current state and its learned parents; this is why we reported exact prediction accuracy. We agree, however, that if the action distribution can be imbalanced. Thus, we will report the F1-score results in the revision.
>
> **Why split explanation and trust metrics across global vs. local models**
>
> Our rationale is that the two explanation objects serve different purposes in the current design. The global graph provides an episode-level overview of dependencies, whereas the local model is state-specific and is the artifact participants used to anticipate what the agent would do next. This is why the explanation-quality items were applied to both graph types, but the trust items were focused on the local model, whose role is closer to action anticipation and context-based reasoning. The paper currently states this too implicitly, and we will make this design rationale explicit in the revision.

---

> > ### Author Rebuttal · Reviewer_Xe6L · 2026-04-02
> >
> > Much of the rebuttal promises fixes, as opposed to describing the fixes, but the answers seem overall convincing. As such, I'll increase my score by 1 to a weak accept.
> >
> > A few points that the rebuttal made me consider that may help in the next revision, wherever it appears:
> > 1. Clarify the intended meaning for "Multi-agent" within the paper (it seems the paper means something like "a centralized controller manipulating multiple entities", as opposed to a distributed controller (as in SMAC).
> > 2. I'd be interested to see some comparison of graphs arising from different settings for conservatism. In particular, which settings seem to result in the "hidden mediator" nodes the rebuttal describes appearing, if ever? Similarly, are there nodes that never appear (i.e., light units), indicating that they are NOT part of the plan, as opposed to a "small" or temporary part of the plan? This type of information would help the reader decide reasonable settings if they adopt the framework the paper follows.

---

> > > ### Author Response · Authors · 2026-04-05
> > >
> > > Thank you for the thoughtful follow-up and increasing the score. We appreciate these suggestions and will address them explicitly in the revision.
> > >
> > > 1. We will clarify that the term “multi-agent” in this paper refers to two or more independently trained RL agents (in MicroRTS, player blue and player red), where each agent/player acts as a centralized controller manipulating multiple entities. In MicroRTS, we interpret directed edges from blue units to red units, or among blue units, as reflecting aspects of the blue agent’s strategy; similarly, edges from red units to blue units, or among red units, reflect aspects of the red agent’s strategy. In PongDuel, each agent controls a single paddle, which is a simpler instance of the same overall setting. We agree that this distinction should be made clearer for readers.
> > >
> > > 2. We agree that comparing graphs under different levels of conservatism would be valuable. In the revision, we will include examples contrasting conservative and more exploratory graphs, especially cases where “hidden mediator” nodes begin to appear. We will also add less conservative examples in which more red-unit nodes become visible, to clarify whether missing nodes are due to thresholding or because they are not salient parts of the learned policy.

---

### Decision · Program_Chairs · 2026-04-30

**Decision:**

Accept (regular)

**Comment:**

CausalXRL proposes a framework for explainable reinforcement learning that combines BART-based causal dependency inference with global and local causal graphs to provide multi-level, interpretable explanations of agent behavior, evaluated on benchmark RL environments and a qualitative expert study in MicroRTS. Three reviewers recommend Weak Accept, recognizing the well-motivated combination of causal reasoning and XRL, the scalability advantage over conditional-independence-test-based alternatives, and the value of the dual-level explanation structure. One reviewer recommends Rejection, primarily on the grounds that the baseline comparisons are insufficient—specifically, the lack of direct comparison against closely related causal-XRL methods limits the strength of the empirical claims. Following AC-facilitated discussion, this disagreement was not fully resolved, and the dissenting reviewer maintained their score.

Having carefully considered the full discussion, the AC judges that the baseline concern, while legitimate and should be addressed in a camera-ready revision, does not constitute a fundamental flaw that disqualifies the contribution. The core technical approach is sound, the authors have committed to expanding baseline comparisons and provided concrete plans for doing so, and the three positive reviewers—including the most expert reviewer on XRL and user studies—find the contribution sufficient. The paper should be revised to: (1) expand causal discovery baselines to include GES and FCI, and add stronger causal-XRL comparisons with explicit justification where direct comparison is infeasible; (2) reframe the user study claims as exploratory and suggestive rather than statistically powered; (3) improve figure clarity and presentation, particularly Figure 2 and the causal edge semantics; and (4) add a sensitivity analysis for the percentile thresholds.